# Exploring the Successions in Microbial Community and Flavor of *Daqu* during Fermentation Produced by Different Pressing Patterns

**DOI:** 10.3390/foods12132603

**Published:** 2023-07-05

**Authors:** Ping Huang, Yao Jin, Mingming Liu, Liqun Peng, Guanrong Yang, Zhi Luo, Dongcai Jiang, Jinsong Zhao, Rongqing Zhou, Chongde Wu

**Affiliations:** 1College of Biomass Science and Engineering, Sichuan University, Chengdu 610065, China; 2021223080042@stu.scu.edu.cn (P.H.); yaojin12@scu.edu.cn (Y.J.); zhourqing@scu.edu.cn (R.Z.); 2Sichuan Liquor Group, Chengdu 610000, China; 18381300889@163.com (M.L.); scniangjiu@sian.com (G.Y.); 3Sichuan Yibin Xufu Liquor Co., Ltd., Yibin 644000, China; xufu9@126.com (L.P.); 13778929757@163.com (Z.L.); jiangdongcaidf@163.com (D.J.)

**Keywords:** volatile compounds, microbial interactions, functional predictions, influencing factors

## Abstract

*Daqu* can be divided into artificially pressed *daqu* (A-*Daqu*) and mechanically pressed *daqu* (M-*Daqu*) based on pressing patterns. Here, we compared the discrepancies in physicochemical properties, volatile metabolites, and microbiota features between A-*Daqu* and M-*Daqu* during fermentation and further investigated the factors causing those differences. A-*Daqu* microbiota was characterized by six genera (e.g., *Bacillus* and *Thermoactinomyces*), while five genera (e.g., *Bacillus* and *Thermomyces*) dominated in M-*Daqu*. The flavor compounds analysis revealed that no obvious difference was observed in the type of esters between the two types of *daqu*, and M-*Daqu* was enriched with more alcohols. The factors related to differences between the two types of *daqu* were five genera (e.g., *Hyphopichia*). The functional prediction of microbial communities revealed that the functional discrepancies between the two types of *daqu* were mainly related to ethanol metabolism and 2,3-butanediol metabolism. This study provided a theoretical basis for understanding the heterogeneity of *daqu* due to the different pressing patterns.

## 1. Introduction

*Daqu* is a kind of microbial fermenting starter, which is rich in a large number of microorganisms and various metabolites, such as various enzymes and flavor substances, with the functions of saccharification, fermentation, and aromatization, and *daqu* plays an important role in the brewing of *baijiu* [1]. *Daqu* is made from wheat as the main raw material or, in some cases, barley or peas [2]. According to the highest temperature reached during production, *daqu* is divided into low, medium, and high temperatures, which are applied to the production of *Qingxiangxing*, *Nongxiangxing*, and *Jiangxiangxing baijiu*, respectively, and contribute unique flavors to *baijiu* [3]. Generally, the flavor and quality of *baijiu* are closely related to the microbial community throughout the production process, especially to the microorganisms in *daqu*, as a high percentage (~20%) of *daqu* is added to the raw grains required for the fermentation of *Nongxiangxing baijiu* [4].

In recent years, the microbial communities in *daqu* have drawn increasing attention due to the advancement of high-throughput sequencing technology, and extensive and in-depth studies have been conducted on the microbial composition of *daqu*. The variations of volatile compounds and the interactions between microorganisms and the environmental factors during the fermentation of *daqu* were investigated using high-throughput, sequencing-based, culture-independent technology combined with culture-dependent methods [5,6]. 

Recently, the demand for *daqu* is increasing quickly, along with the consumer market for *baijiu*. The cumulative production of *Nongxiangxing baijiu* in China reached 1.86 million kiloliters in May 2023 (data from National Bureau of the Statistics of the People’s Republic of China). *Daqu* manufacturers currently prefer to make *daqu* using a mechanical pattern due to its higher efficiency compared with artificial manufacturing, with which it is difficult to meet the demands for *daqu* output [7]. Manufacturers have started using automated facilities for bricks, decreasing labor intensity by more than half [8]. The traditional preparation process of *Nongxiangxing daqu* consists of three main stages: (i) wheat infiltration and grinding; (ii) shaping into bricks; and (iii) fermentation and storage. The shaping stage is the pressing of the raw material into bricks. In this stage, two types of brick forming can be used, i.e., an artificial *daqu*-stepping mold or a pressed-shaping machine, but with the same production process (e.g., raw materials, process parameters) [9]. The mechanically pressed *daqu* bricks are M-*Daqu*, while the artificially pressed daqu are A-*Daqu*. Although the process principle of M-*Daqu* is the same as that of A-*Daqu*, there are still differences in quality and microbial community structure [10,11]. The investigation of the differences between M-*Daqu* and A-*Daqu* will help to promote the fully automatic production of *Nongxiangxing daqu*. Currently, numerous scholars have carried out far-reaching research on *daqu* during the fermentation process, revealing the microbial community succession and the correlation between microorganisms and flavor substances, and they have investigated the impact of environmental factors on *daqu* microbial communities [5,6]. However, the differences in microbial community structure and flavor between M-*Daqu* and A-*Daqu* during fermentation remain unclear. Therefore, in this study, the succession of microbial communities and volatile compounds of the two types of *daqu* during fermentation were investigated. Then, the biomarkers that distinguish the two types of *daqu* were revealed. Random forest (RF) analysis was used to identify the main factors influencing the changes in microbial community and physicochemical properties of *daqu* during fermentation. On this basis, the function of microorganisms in flavor development was further revealed by applying the PICRUSt2 tool based on 16S rRNA and ITS sequencing results. The results presented in this study may contribute to promoting the technological development for *daqu* manufacturing and improve the quality of *daqu* and Chinese *baijiu*. 

## 2. Materials and Methods

### 2.1. Collection and Preparation of Sample

Samples of artificially pressed *daqu* (A-*Daqu*) and mechanically pressed *daqu* (M-*Daqu*) were obtained from Sichuan Yibin Xufu Liquor Co., Ltd. (Yibin, Sichuan, China), a representative liquor enterprise located in Sichuan province, China (105°36′47.4″ E, 28°45′55.9″ N). A-*Daqu* (25 cm × 24 cm × 5 cm) adopted an artificially pressed method, while M-*Daqu* (25 cm × 24 cm × 5 cm) adopted a mechanical pressed method. In September 2021, three parallel samples were randomly chosen from the upper, middle, and bottom stacked layers in the Qu room (incubating room). The A-*Daqu* samples were harvested on the 5th, 10th, 15th, 25th, and 40th days during the *daqu* incubation process, and they were labeled as A5, A10, A15, A25, and A40, respectively. On the 5th, 10th, 15th, 20th, and 30th days of fermentation, M-*Daqu* was collected, and these samples were marked as M5, M10, M15, M20, and M30, respectively. The initial raw material mixtures were labeled D0. The 33 samples were collected, ground, and mixed separately, with the amount of powder required for DNA extraction collected and stored separately in a −80 °C refrigerator, and the remainder was sealed in sterile bags and maintained at −80 °C until further analysis.

### 2.2. Determination of Physicochemical Properties and Enzymatic Activities

A gravimetric approach was employed to evaluate the water content of the samples, which dried the samples to a constant weight at 105 °C (QB/T 4257–2011) [12]. The total titratable acid was measured using the direct titration method with 0.1 mol/L NaOH solution to the endpoint of a pH of 8.2. Fermenting power (FP), saccharification power (SP), esterifying power (EP), and liquefying power (LP) were determined based on national professional standard techniques (QB/T 4257–2011). The fermentation power was expressed as the weight of carbon dioxide produced from fermentable sugars in 0.5 g of *daqu* at 30 °C for 72 h. The saccharification power was defined as the production of glucose converted from soluble starch in 1 g of *daqu* at 35 °C and a pH of 4.6 in 1 h. The esterifying power was expressed as the production of ethyl hexanoate synthesized from hexanoic acid and ethanol in 25 g of *daqu* at 35 °C for 7 d. One unit of liquefying power was defined as the amount of soluble starch that can be liquefied in 1 g of absolute dry *daqu* in 1 h. All physicochemical and enzymatic parameters were determined in triplicate.

### 2.3. Analysis of Volatile Metabolites by HS-SPME-GC/MS

Volatile chemicals were extracted with a headspace-solid phase microextraction (HS-SPME) technique, applying a DVB/CAR/PDMS fiber 50/30 μm three-phase extraction head (Supelco, Inc., Bellefonte, PA, USA). Briefly, 0.5 g of sample was placed in a 15 mL headspace vial with 10 µL 2-octanol (0.0158 g/100 mL) dissolved in chromatographic grade methanol as an internal standard. After balancing the head bottle in a warm sink for 15 min at a constant temperature of 60 °C under stirring, the fiber was pushed into the bottle and extracted for 50 min at the same temperature. Immediately thereafter, the fiber was inserted into the gas chromatograph-mass spectrometer (GC-MS) input, and the analyte was thermally desorbed for 3 min at 250 °C. Volatile chemicals were determined by a Shimadzu (Japan) GCMS-QP2010SE machine equipped with a DB-WAX capillary column (60 m × 0.32 mm × 0.25 μm). The desorption duration was 3 min, and the input temperature was 250 °C. The split ratio was 20:1 with a constant flow rate of 1.0 mL/min of helium carrier gas. The temperature protocol was as follows: 40 °C for 5 min, then increased to 220 °C at 5 °C/min and maintained for 5 min. The ion source temperature, interface temperature, EI ionization mode, and mass range were all adjusted to 250 °C, 200 °C, 70 ev, and 40–500 *m*/*z*, respectively, for MS parameters. The compounds were identified by comparing mass spectra to the NIST05 spectral database.

### 2.4. DNA Extraction, PCR Amplification and Illumina MiSeq Sequencing

Microbial community genomic DNA was extracted from 33 *daqu* samples using the E.Z.N.A.^®^ soil DNA Kit (Omega Bio-tek, Norcross, GA, USA) according to the manufacturer’s instructions. The DNA extract was checked on 1% agarose gel, and the DNA concentration and purity were determined with a NanoDrop 2000 UV-vis spectrophotometer (Thermo Scientific, WA, USA). The hypervariable region V3-V4 of the bacterial 16S rRNA gene and the ITS regions of fungal rRNA gene were amplified with the primer pairs 338F/806R and ITS1F/ITS2R, respectively, by an ABI GeneAmp^®^ 9700 PCR thermocycler (ABI, San Francisco, CA, USA). The PCR amplification of 16S rRNA genes was performed as follows: initial denaturation at 95 °C for 3 min, followed by 29 cycles of denaturing at 95 °C for 30 s, annealing at 53 °C for 30 s and extension at 72 °C for 45 s, a single extension at 72 °C for 10 min, and ending at 10 °C. The PCR amplification of ITS regions of fungal genes were performed as follows: initial denaturation at 95 °C for 3 min, followed by 35 cycles of denaturing at 95 °C for 30 s, annealing at 55 °C for 30 s and extension at 72 °C for 45 s, a single extension at 72 °C for 10 min, and ending at 10 °C. The PCR mixtures contained 5 × TransStart FastPfu buffer 4 μL, 2.5 mM dNTPs 2 μL, forward primer (5 μM) 0.8 μL, reverse primer (5 μM) 0.8 μL, TransStart FastPfu DNA Polymerase 0.4 μL, BAS 0.2 μL, template DNA 10 ng, and finally ddH_2_O up to 20 μL. PCR reactions were performed in triplicate. The PCR products were extracted from 2% agarose gel and purified using the AxyPrep DNA Gel Extraction Kit (Axygen Biosciences, Union City, CA, USA) according to the manufacturer’s instructions and quantified using a Quantus™ Fluorometer (Promega, Madison, WI, USA). 

Purified amplicons were pooled in equimolar and paired-end sequenced on an Illumina MiSeq PE300 platform/NovaSeq PE250 platform (Illumina, San Diego, CA, USA) according to the standard protocols by Majorbio Bio-Pharm Technology Co., Ltd. (Shanghai, China).

### 2.5. Processing of Sequencing Data

The raw paired-end reads were demultiplexed, quality-filtered by fastp version 0.20.0 [13], and merged by FLASH version 1.2.7 [14] with the following criteria. (i) The 300-bp reads were truncated at any site receiving an average quality score of <20 over a 50-bp sliding window, the truncated reads shorter than 50 bp were discarded, and reads containing ambiguous characters were also discarded. (ii) Only overlapping sequences longer than 10 bp were assembled according to their overlapped sequences. The maximum mismatch ratio of overlap region was 0.2. Reads that could not be assembled were discarded. (iii) Regarding exact matching of barcodes, primers were allowed two mismatches, and reads containing ambiguous characters are removed; operational taxonomic units (OTUs) with 97% similarity cutoff [15,16] were clustered using UPARSE version 7.1 [15], and chimeric sequences were identified and removed. The taxonomy of each OTU representative sequence was analyzed by RDP Classifier version 2.2 [17] against the 16S rRNA database Silva (Release138) an ITS rRNA database (unite 8.0) using a confidence threshold of 0.7.

### 2.6. Statistical Analysis and Visualization

The indices of α diversity, ACE, Chao1, and Shannon indices, were calculated using mothur (version v.1.30.2). Venn diagrams were constructed to visualize the shared and unique volatile metabolites among *daqu* samples using a free online website (http://www.ehbio.com/test/venn/#/ (accessed on 2 July 2023)). Heatmaps illustrating the variation of gene abundances were produced using TBtools 0.665. β diversity was analyzed using principal coordinate analysis (PCoA) on the basis of the UniFrac distance using an unweighted algorithm. Biomarkers were predicted in the two types of *daqu* with R (version 3.3.1) using the randomForest package. Co-occurrence network visualization between microorganisms was performed using the Gephi program (version 0.9.2). Spearman’s rank correlation analysis was performed using the R package Hmisc to investigate the interactions between the main microbial genera and the main volatile compounds. Using the R vegan package, redundancy analysis was used to investigate the relationship of samples with physicochemical indices and enzymatic indicators [18]. The rfPermute package was used to assess the main predictors of changes in the physicochemical properties of the two types of *daqu* [19]. Phylogenetic Investigation of Communities by Reconstruction of Unobserved States II (PICRUST II) was used to predict the microbial community function and pathway enrichment of metabolites based on the Kyoto Encyclopedia of Genes and Genomes (KEGG) database [20]. Bubble plots demonstrating the interactions between the key functional enzymes and taxa were generated using R, with ggplot2.

## 3. Results

### 3.1. Temporal Changes in Physicochemical Characteristics and Enzymatic Activities of Daqu during Fermentation

The changes in the physicochemical properties and enzymatic activities of the two types of *daqu* during fermentation were investigated (Appendix A). The water content of *daqu* gradually decreased with the extension of fermentation time and stabilized after 20 days in both types of *daqu* samples (Appendix A). As for acidity, it increased throughout the fermentation period in both types of *daqu*. (Appendix A). Regarding fermenting power (FP), similar change trends for both *daqu* samples were observed during fermentation, which increased to the maximum on the fifth day and then decreased gradually (Appendix A). Analysis of the saccharifying power (SP) of *daqu* showed that SP in A-*Daqu* grew to its peak value (1006 mg/g·h) on the 10th day and subsequently decreased to its minimum (610 mg/g·h) on the 25th day. M-*Daqu* showed a similar trend of changes in saccharifying power. However, the saccharifying power of A-*Daqu* was higher than that of M-*Daqu* (Appendix A). The esterifying power (EP) of the two types of *daqu* differed significantly, with A-*Daqu* fluctuating up and down during the fermentation phase, whereas M-*Daqu* had a steady upward tendency (38.24–282.58 mg/25 g·7 d) (Appendix A). In the first 10 days, the liquefying power (LP) of A-*Daqu* swiftly rose to the maximum (0.086 g/g·h) and then gradually declined to its lowest point (0.017 g/g·h) on the 40th day. The liquefying power of M-*Daqu* displayed a consistent upward trend (0.0044–0.028 g/g·h) with a relatively small variation (Appendix A).

### 3.2. Analysis of Volatile Compounds

The flavor compounds of the two types of *daqu* samples at various fermentation stages were identified (Appendix A). As shown in Appendix A, a total of 97 compounds were detected, comprising 59 esters, 16 alcohols, 7 ketones and 8 aldehydes, 4 acids, 1 aromatic, 1 phenol, and 1 pyrazine. The amount and content of esters in both types of *daqu* during fermentation accounted for the majority of all flavor substances (Figure 1A,B).

Among the semi-quantified esters, hexadecanoic acid methyl ester, hexanoic acid methyl ester, octanoic acid methyl ester, and hexanoic acid ethyl ester were the dominant esters (Appendix A). The Upset diagram shows the number of flavor substances common and specific to each sample, and these specific flavor substances are shown in the Venn networks (Figure 1C,D). There were 21 distinctive flavor substances in A-*Daqu*, and the contents of which were significantly higher than those in M-*Daqu* (15). In addition, PCoA showed that the flavor substances could be broadly divided into three groups: A, M, and D0 (*p* < 0.05). LEfSe analysis was applied to further identify statistically representative flavor substances, with alcohols being enriched in M-*Daqu* and aldehydes, ketones, and phenols being enriched in D0 (Figure 1F,G). In addition, it was noteworthy that 2,3-butanediol and 2,3-butanediol [R-(R*, R*)] occurred almost exclusively in large quantities at the end of fermentation in M-*Daqu*. The characteristic volatiles in A-*Daqu* were hexanoic acid ethyl ester, hexadecanoic acid ethyl ester, etc. Hexanoic acid ethyl ester, an important flavor substance in *Nongxiangxing baijiu*, was also detected in the two types of *daqu* at high levels, with the greatest amount of hexanoic acid ethyl ester occurring at the end of fermentation.

### 3.3. The Succession of Microbial Community in the Two Types of Daqu

High-throughput sequencing was employed to investigate the succession of microbial community in the two types of *daqu.* The 33 samples had a range of 32,331–57,908 effective sequences for bacteria and 42,495–90,377 for fungi, respectively (Appendix A). The saturation plateau of the rarefaction curves based on OTU numbers revealed that the sequencing depth in our study was suitable to represent the community structure of samples (Appendix A). The total number of shared bacterial genera in all samples was 34, of which 124 genera were unique to A25 (Appendix A). *Vulgatubacter* and *Gluconobacter*, for example, were genera unique to M-*Daqu*, while *Aggregatibacter, Trichococcus*, etc., were exclusive genera in A-*Daqu* (Appendix A). Appendix A demonstrates that there were 18 fungal genera shared by M-*Daqu* and A-*Daqu*. Species richness was measured using the ACE and Chao1 indices, and species diversity was measured using the Simpson and Shannon indices (Appendix A). The lowest bacterial diversity was observed in M30, whereas the maximum was reported in A25. The lowest level of fungal community diversity was discovered in M20, while the most was detected in A5. 

The results from temporal profiling of the bacterial community structure of A-*Daqu* and M-*Daqu* during fermentation are shown in Figure 2. The top 6 bacterial phyla with an abundance greater than 1% were demonstrated in A-*Daqu* and M-*Daqu* (Figure 2A). Both Firmicute and Proteobacteria were the dominant phyla in A-*Daqu* and M-*Daqu*. As for fungi, three phyla were detected in A-*Daqu* and M-*Daqu,* including Ascomycota, Basidiomycota, and Mucoromycota (Figure 2B). Ascomycota dominated the fungal phyla in both A-*Daqu* and M-*Daqu*. On the whole, both A-*Daqu* and M-*Daqu* maintained a relatively stable fungal community structure at the phylum level compared to that of bacteria during the fermentation process. 

At the genus level, the most predominant bacterial genus in both A-*Daqu* and M-*Daqu* was *Bacillus* (Figure 2C). In A-*Daqu*, the relative abundance of *Bacillus* increased sharply to 75.37% on the fifth day, followed by a decrease to 9.31% as fermentation continued. The amounts of *Thermoactinomyces*, *Weissella*, and *Lactobacillus* all rose during the late stage of fermentation (15–40 d). In M-*Daqu*, *Bacillus* rose to the maximum value (60.97%) at day 10, then fell to a lesser amount (6.71% for M15), and ultimately returned to a higher value (46.52%) at day 30. *Streptococcus*, and *Thermoactinomyces* were discovered in A-*Daqu* and rarely in M-*Daqu*. However, *Pantoea* and *Corynebacterium* were more prevalent in M-*Daqu* than in A-*Daqu*. 

The most predominant fungal genera in A-*Daqu* and M-*Daqu* were Unclassified_f_Dipodascaceae and *Thermomyces*, respectively (Figure 2D). The relative abundance of *Thermomyces* was at a low level until A40 (76.87%). In M-*Daqu*, the relative abundance of *Thermomyces* peaked at 85.53% in M10 and then dropped to 28.5% in M30. Compared to *Thermomyces,* the relative abundance of *Thermoascus* showed the reverse tendency.

### 3.4. Divergence of Microbial Communities in Two Types of Daqu Samples

Principal coordinate analysis (PCoA) was performed using the unweighted algorithm based on the genus level to evaluate beta-diversity (β-diversity) in the microbial communities of the A-*Daqu* and M-*Daqu* samples (Figure 3). The results demonstrated that the bacterial community profiles could be divided into A-*Daqu* and M-*Daqu* groups during fermentation (Figure 3A). As demonstrated in Figure 3A, bacterial communities in M-*Daqu* samples were more aggregated during the fermentation process, while in A-*Daqu,* they were more dispersed. As for fungi, fungal community profiles might also be divided into A-*Daqu* and M-*Daqu* groups (Figure 3B). Compared to bacterial communities, the fungal communities in both the A-*Daqu* and M-*Daqu* groups were more concentrated. In addition, the microbial community of D0 was highly separate from that in other samples, and the composition was clearly distinguishable, consistent with previous studies [21]. 

The results of PCoA showed that the microbial communities of the two types of *daqu* during the fermentation process differed significantly and could be clearly distinguished as A-*Daqu* and M-*Daqu*. A random forest learning algorithm was used to regress the relative abundance of each microbial genus with respect to the corresponding *daqu*, and the top eight bacterial genera and top eight fungal genera were obtained as biomarkers to distinguish A-*Daqu* from M-*Daqu* based on importance values > 2 of the microbial genera (Figure 3C,D). Among them, bacterial genera belonged to Proteobacteria and Firmicutes (Figure 3E), while the fungal genera were classified as Basidiomycota and Ascomycota (Figure 3F). Among the two types of *daqu*, the relative abundance of *Pantoea*, *Kosakonia,* and *Staphylococcus* was higher in M-*Daqu*, and conversely, A-*Daqu* had greater relative abundances of *Acinetobacter*, *Streptococcus*, and *Thermoactinomyces*. On the other hand, the relative abundance of all seven genera, except *Thermomyces*, was lower in M-*Daqu* than that of the corresponding genera in A-*Daqu*.

### 3.5. Interaction Network between Microbial Communities and Volatile Compounds in Daqu

Spearman’s correlation analysis (*p* < 0.05) between the top 20 volatiles and dominant microorganisms, including the top 10 fungal and top 10 bacterial genera in relative abundance, was conducted, aiming to clarify their relationship and to obtain more meaningful information (Figure 4). In A-*Daqu*, *Lactobacillus* and *Theromactinomyces* were positively correlated with almost all the flavor substances (Figure 4A). *Pantoea*, *Weissella*, *Saccharopolyspora,* and *Lactobacillus* were positively correlated with hexanoic acid ethyl ester and hexanoic acid butyl ester. The bacteria *Pantoea*, *Weissella*, *Saccharopolyspora*, and *Thermomyces* were positively connected to hexanoic acid, an exclusive flavor component in A-*Daqu*. In M-*Daqu*, *Aspergillus* and *Thermoascus* were positively correlated with almost all flavor substances (Figure 4B). 2,3-butanediol, [R-R*, R*] was a flavor substance distinctive to M-*Daqu*, which was positively correlated with *Aspergillus* and *Thermoascus.*

### 3.6. Microbial Interactions and Correlation between Microbial Communities and Physicochemical Properties

To elucidate the interactions between microbial communities during the fermentation of the two types of *daqu*, we explored the co-occurrence and co-exclusion patterns of microbial communities based on Spearman’s rank correlation (|ρ| > 0.6 and *p* < 0.05) (Figure 5A,B). A total of 50 nodes and 117 edges were obtained in the 50 dominant genera, including the top 25 fungal and top 25 bacterial genera in relative abundance in A-*Daqu* and 50 nodes and 184 edges in the 50 dominant genera in M-*Daqu*. In A-*Daqu*, microbial interactions were overwhelmingly positively correlated (99.15%), except for *Bacillus* and *Lactococcus,* which showed a negative correlation (Figure 5A). In M-*Daqu*, positive correlations reached 96.2%, with *Bacillus* showing negative correlations with *Pseudomonas* and *Lactococcus* showing negative correlations with *Pantoea*, different from the reciprocal relationships presented by *Bacillus* in A-*Daqu* (Figure 5B). Comparing the two types of *daqu*, there were more edges of microbial networks in M-*Daqu* than in A-*Daqu*, indicating that the interactions between microorganisms in M-*Daqu* were closer.

The correlations between six physicochemical properties and the top 15 bacteria and top 15 fungi in relative abundance were investigated by redundancy analysis (Figure 5C,D). *Lactobacillus*, *Weissella*, and *Leucanostoc* were positively correlated with acidity (Figure 5C). *Bacillus*, *Streptococcus*, *Hyphopichia*, and *Wickerhamomyces* were positively correlated with FP, SP, EP, and LP (Figure 5D). In addition, the results also revealed that LP and acidity were the key determinants representing the changes in microbial taxa during fermentation of A-*Daqu*, whereas M-*Daqu* was determined by water content.

### 3.7. Factors Contributing to the Differences in the Microbial Communities and Physicochemical Properties of the Two Daqu

The Mantel test was employed to further investigate the relationship between biomarkers and other microorganisms in the microbial community and to clarify the factors responsible for differences in the microbial community in the two types of *daqu* during fermentation. As shown in Figure 5E, the bacterial biomarkers were correlated with *Bacillus* (r = 0.2, *p* < 0.01), *Pediococcus* (r = 0.21, *p* < 0.01), and *Fusobacterium* (r = 0.27, *p* < 0.01). On the other hand, the fungal biomarkers were correlated with *Bacillus* (r = 0.17, *p* < 0.01), *Pediococcus* (r = 0.22, *p* < 0.01), and *Kocuria* (r = 0.29, *p* < 0.01). Figure 5F suggested that the bacterial biomarkers were correlated with *Thermoascus* (r = 0.28, *p* < 0.01), *Fusarium* (r = 0.2, *p* < 0.01), *Wallemia* (r = 0.23, *p* < 0.01), and *Diutina* (r = 0.24, *p* < 0.01) Furthermore, the fungal biomarkers were correlated with *Fusarium* (r = 0.4, *p* < 0.01), *Alternaria* (r = 0.31, *p* < 0.01), *Epicoccum* (r = 0.36, *p* < 0.01), and *Wallemia* (r = 0.36 *p* < 0.01). The results showed that both fungi and bacteria were almost always significantly associated with fungal and bacterial biomarkers. Therefore, we hypothesized interactions between biomarkers and other microorganisms, and the results indicated that microorganisms interacted with biomarkers and influenced the abundance of biomarkers, thus leading to the differences in the microbial community structures in the two types of *daqu*. 

Differences in microbial communities may cause discrepancies in the physicochemical properties between the two types of *daqu*, so PCoA was used to analyze the disparities in physicochemical properties between the two types of *daqu*. The results showed that there were significant differences in the physicochemical properties (Appendix A). Therefore, we further investigated the factors responsible for the differences in physicochemical properties. Spearman’s correlation test was used to investigate the relationship between each biomarker and the physicochemical properties (Figure 5G,H). Most of the bacterial and fungal biomarkers were significantly correlated with these properties. In addition, random forest analysis was used to identify the main factors influencing the changes in the physicochemical properties of *daqu* during fermentation (Figure 6). In A-*Daqu* (Figure 6A,B), *Thermoactinomyces*, *Thermomyces*, and *Hyphopichia* were important factors in explaining water content variation, while *Aquabacterium* and *Hyphopichia* were important factors in explaining the changes in acidity and EP. *Streptococcus* and *Apiotrichum* had high MSE values and played major roles in explaining FP. *Pantoea* and *Hyhopichia* were important factors that explained the change in LP. Finally, *Streptococcus* and *Hyphopichia* were important factors in explaining SP. As for M-*Daqu* (Figure 6C,D), *Thermoactinomyces*, *Aquabacterium*, *Thermomyces*, and *Apiotrichum* were essential factors in explaining water content, acidity, and EP. *Thermomyces* and *Apiotrichum* were factors important in explaining FP. *Kosakonia*, *Streptococcus,* and *Hyphopichia* were important factors in explaining SP. Thus, the variations in physicochemical properties in the two types of *daqu* were mainly caused by *Thermoactinomyces*, *Aquabacterium*, *Streptococcus*, *Thermomyces*, and *Hyphopichia*.

### 3.8. Prediction of the Microbial Functions in Daqu

The results of PICRUSt2 based on 16S rRNA and ITS sequencing data were used to predict the interconnections between the two types of *daqu* and specific functional enzymes during fermentation. We selected 64 enzymes from the KEGG database that were potentially involved in substrate degradation and flavor formation in the two types of *daqu* and then categorized them into 16 functional components (Figure 7A). The abundance of enzymes related to starch catabolism, ethanol metabolism, and ethanol aldehyde-carboxylate metabolism were high during the fermentation process in both types of *daqu*. In addition, the relative abundance of these enzymes in M-*Daqu* was generally higher than that in A-*Daqu*, and these enzymes were almost exclusively secreted by the fungi. Among them, the glucan 1,4-alpha-glucosidase (EC 3.2.1.3), the dominant enzyme related to starch catabolism, had the highest abundance in both types of *daqu.* Starch was broken down into glucose by the action of enzymes (EC 3.2.1.3, EC 3.2.1.133, EC 3.2.1.1) and further metabolized via the glycolytic pathway to produce pyruvate, which was then converted as a precursor substance to form organic acids (Figure 7B). Additionally, alcohol dehydrogenase (EC 1.1.1.1), which is the main enzyme involved in the metabolism of ethanol and ethanol aldehyde-carboxylate, converts pyruvate into acetaldehyde and ethanol in the absence of oxygen. As shown in Appendix A, the contents of 2,3-butanediol and other higher alcohols in M-*Daqu* were higher than those in A-*Daqu*. It is noteworthy that the abundance of enzymes (EC 2.2.2.6, EC 4.1.1.5, EC 1.1.1.76, EC 3.1.1.1, EC1.1.1.1, and so on) associated with 2,3-butanedione biosynthesis and higher alcohol formation were indeed greater in M-*Daqu* than in A-*Daqu* (Figure 7).

## 4. Discussion

This study compared the microbial community structure, flavor, and microbial functions between two different types of Nongxiangxing *daqu*, and it revealed the factors that caused differences in microbial communities and physicochemical properties between the two types of *daqu*. Then, the microbial interactions and functions of *daqu* were elucidated based on high throughput sequencing. 

The physicochemical characteristics of *daqu* are commonly used as a benchmark for evaluating the quality of *daqu*, and variations in the physicochemical properties are related to the microbial composition of *daqu*. The acidity of the two types of *daqu* increased with time during fermentation, and similar results were also reported by Guan et al. (2019) [9]. At the same time, the relative abundance of *Lactobacillus* increased with time (Figure 2C,D). In addition, RDA results also showed that the relationship between acidity and *Lactobacillus*, *Weissella*, and *Leucanostoc* was positively correlated with COS > 0 (Figure 5C). The fermenting power of *daqu* was mainly positively correlated with *Wickerhamomyces* and *Leuconostoc* (Figure 5C,D), consistent with previous studies [9,22] The sudden increase in saccharifying power after 20 and 25 days of fermentation for M-*Daqu* and A-*Daqu*, respectively, may be ascribed to the increased relative abundance of the glycosylase-producing strain *Thermoascus* at the corresponding times [23] (Figure 2). Liquefying power reflects the ability of *daqu* to decompose starch, and *Bacillus* was reported to be the main contributor to starch degradation by secreting starch degrading enzymes, such as alpha-amylase [24]. Meanwhile, samples A10 and M10 showed higher liquefying power and higher relative abundance of *Bacillus* compared to other *daqu* samples (Appendix A and Figure 2). RDA also showed that *Bacillus* was positively correlated with saccharifying power and liquefying power (Figure 5C), consistent with Shi et al. (2021) [25].

To further investigate the difference in microbial community in two types of *daqu* during fermentation, combined Illumina MiSeq sequencing and bioinformatic analysis was performed. The results demonstrated that Firmicutes and Proteobacteria were the dominant bacterial phyla in both A-*Daqu* and M-*Daqu*, and similar results were also reported by previous research [22,26].The dominant bacterial genera identified in both types of *daqu* in this study were *Bacillus*, *Lactobacillus,* and *Weissella*, which were often detected in previous research in high- and medium-temperature *daqu* [27,28]. In addition, *Saccharopolyspora* and *Thermoactinomyces* were the dominant bacterial genera in A-*Daqu*, and *Pantoea*, *Staphylococcus*, *Leuconostoc,* and *Pediococcus* were the dominant bacterial genera in M-*Daqu*. *Thermoactinomyces* was regarded as a biomarker and a facilitator of changes in physicochemical properties in this study (Figure 3C and Figure 6). In addition, it was suggested that *Thermoactinomyces* may play important roles in maintaining flavor diversity and community balance [29]. *Pantoea* and *Staphylococcus*, which served as biomarkers to distinguish A-*Daqu* from M-*Daqu* in this study (Figure 3C), were susceptible to carbon dioxide, water content, and acidity [30]. Of these genera, *Pantoea* was almost exclusively present in M-*Daqu* and had the highest relative abundance on the fifth day of fermentation (Figure 2C). Previous research has indicated that *Pantoea* was present in large quantities only in the early stages of *daqu* fermentation [31], and it was reported to be involved in lipid synthesis during the fermentation of *daqu*, producing low-molecular weight lipopolysaccharides [32,33]. *Pantoea* promotes the conversion of sulfur-containing compounds into less volatile sulfur-containing amino acids [34].The *Staphylococcus* in *daqu* may result from the raw material wheat of *daqu* [35]. It was inferred that the *Staphylococcus* may mainly originate from raw wheat and may be affected by the microenvironment in M-*Daqu* because of inconsistencies in the looseness of M-*Daqu* compared with A-*Daqu*, resulting in inconsistent succession patterns of *Staphylococcus* in the two types of *daqu*. In addition, the formation of aromatic chemicals by *Staphylococcus*, including 3-methyl-1-butanol, and diacetyl,2-butanone, may be crucial in the brewing of *baijiu* [36]. 

In this study, the dominant eukaryotic microorganisms identified in both types of *daqu* were *Thermomyces*, and *Thermoascus*, which are also considered to be the central microorganisms in the synthesis of ethyl caproate during *baijiu* brewing [5,37]. Among them, *Thermoascus* is also a biomarker of *Nongxiangxing Daqu* in Sichaun [38].In addition, four dominant eukaryotes were found in A-*Daqu* samples, namely *Hyphopichia*, *Wickerhamomyces*, and *Saccharomycopsis*. *Aspergillus* and *Rhizopus* were the dominant eukaryotic microorganisms in M-*Daqu*. *Aspergillus* is an important functional fungal genus in *daqu* due to its ability to secrete acid- and ethanol-resistant extracellular enzymes, such as saccharifying enzyme [39,40,41]. *Rhizopus* can produce lipase and amylase, which can break down protein and starch into sugar and amino acids and contribute particularly to the flavor of Chinese *baijiu* [42]. Figure 2 demonstrates that the relative abundance of *Hyphopichia* in A-*Daqu* was high during the first 10 days of fermentation, which may be related to its thermal stability [43]. When A-*Daqu* were pressed, more microorganisms in air, such as *Saccharomycopsi,* remain in the *daqu* due to multiple stepping. This fact may be why *Saccharomycopsis* was the dominant genus in A-*Daqu* rather than M-*Daqu*. In addition, Du et al. (2019) [31] suggested that *Saccharomycopsis* in *daqu* may originate from the air.

The types and contents of volatiles in the two types of *daqu* increased with fermentation time, consistent with previous studies [5]. Spearman’s correlation analysis indicated that ester compounds were positively correlated with lactic acid bacteria in A-*Daqu* (Figure 4). Jin et al. (2019) [5] suggested that Lactobacillales (*Enterococcus*, *Pediococcus*, *Lactobacillus*) was also involved in the formation of methyl and ethyl esters. In the case of M-*Daqu*, the esters were positively correlated with *Aspergillus*, consistent with previous studies [22]. Ketones were mainly detected in A-*Daqu*, with 2-octanone being the most abundant (Appendix A). Benzaldehyde dominated the aldehydes, and it was only found in the M-*Daqu* and was positively correlated with *Thermoascus* and *Weissella* (Figure 4B). Moreover, there were significant differences in the alcohols, particularly 2,3-butanediol and 2,3-butanediol, [R-(R*, R*)], which occurred almost exclusively in the later stages of M-*Daqu* fermentation. In general, *Weissella*, *Bacillus*, *Lactobacillus,* and *Staphylococcus* were the main contributors to the production of 2,3-butanediol [29]. In addition, 2,3-butanediol can also be considered one of the biomarkers reflecting the growth of *Bacillus* [37]. However, in this study, in addition to *Bacillus* in M-*Daqu*, which was positively correlated with 2,3-butanediol, *Aspergillus* was also positively correlated with 2,3-butanediol (Figure 4B). The abundance of enzymes involved in the synthesis of 2,3-butanediol was significantly higher in M-*Daqu* than that in A-*Daqu* (EC 2.2.1.6, EC 4.1.1.5 and EC 1.1.1.76), probably due to the higher abundance of *Bacillus* and *Aspergillus* (Figure 7A). Functional predictions of the enzymes in *daqu* showed that the main enzymes in the two types of *daqu* were glucan 1,4-α-glucosidase (EC 3.2.1.3), alcohol dehydrogenase (EC 1.1.1), and β-glucosidase (EC 3.2.1.21). In addition, the abundance of these enzymes was higher in M-*Daqu* than in A-*Daqu*, which was inconsistent with the enzyme activities measured in Appendix A. This finding may be ascribed to the inactivation of enzyme-producing microbes during fermentation, with the DNA still retained in the samples [29]. 

## 5. Conclusions

In summary, the physicochemical characteristics, microbial community structure, and volatiles of the two types of *Nongxiangxing daqu* were investigated during the fermentation process. The results showed that the different production methods led to heterogeneity in the physicochemical properties, microbial community structure, and flavor metabolites of the *daqu*. During fermentation, the enzyme activities of A-*Daqu* were generally higher than those of M-*Daqu*. The predominant bacterial and fungal genera in A-*Daqu* were *Saccharopolyspora*, *Streptococcus*, *Hyphopichia*, *Wickerhamomyces*, *Saccharomycopsis,* and so on, while *Pantoea*, *Staphylococcus*, *Leuconostoc*, *Aspergillus,* and *Rhizopus* predominated in M-*Daqu*. Among these microbes, *Pantoea*, *Staphylococcus*, *Thermoactinomyces,* and *Thermomyces* were the biomarkers with highly important values to distinguish the two types of *daqu*. *Thermoactinomyces*, *Aquabacterium*, *Streptococcus*, *Thermomyces*, and *Hyphopichia* were the main factors contributing to differences in physicochemical properties between the two types of *daqu*. However, the factors that contributed to microbial differences were complex, and further investigation is needed. Benzaldehyde and 2,3-butanediol, [R-(R*, R*)] were only detected in large amounts in M-*Daqu*. The bacterial community in M-*Daqu* showed higher potential for 2,3-butanediol and 2,3-butanediol [R-(R*, R*)] synthesis. Hexanoic acid, phenylethyl alcohol, and pyrazine tetramethyl, on the other hand, were only present in A-*Daqu*. In conclusion, our work may contribute to understand the impacts of different pressing patterns on *daqu* as well as the distinctions between the two types of *daqu*. Additionally, it helps to screen functional microorganisms from A-*Daqu* and apply them to M-*Daqu* to reduce the quality differences between the two types of *daqu*, thus improving the quality of M-*Daqu* and providing a scientific basis for the optimization of the *daqu* production process.

## Figures and Tables

**Figure 1 foods-12-02603-f001:**
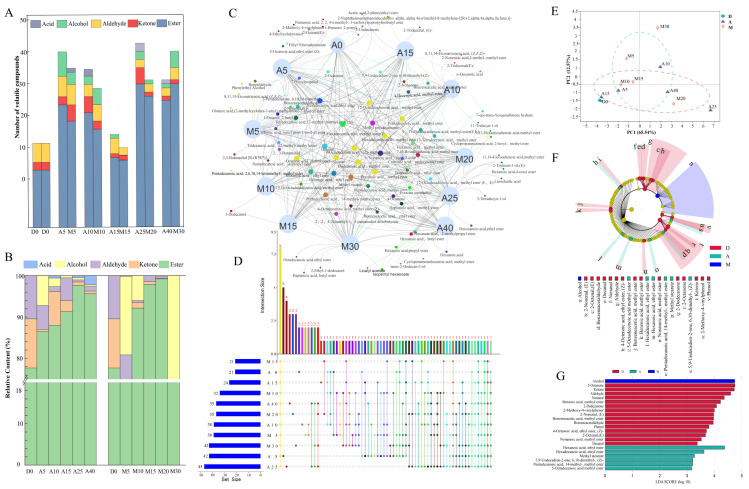
Comparison of volatile compounds of the two types of *daqu* samples based on the results of HS−SPME−GC/MS. (**A**) Variations in the category and number of volatiles. (**B**) Variations in the relative content of each category. (**C**) Shared and unique volatile compounds presented by Venn networks. The color of the circles in plot C corresponds to the color of the bars in plot D, reflecting the shared and unique flavors in the sample. (**D**) The number of shared and unique volatile compounds presented by Upset diagram. (**E**) Variations in volatile compounds based on principal component analysis of different *daqu* samples. (**F**,**G**) The cladogram and characteristic volatiles of different *daqu* samples obtained from LEfSe analysis. The red, blue and green nodes indicate microbial taxa that are significantly enriched in D0, A and M groups, respectively, and have a significant effect on the difference between groups. the light yellow nodes indicate microbial taxa that are not significantly different in any of the different groups, or have no significant effect on the difference between groups.

**Figure 2 foods-12-02603-f002:**
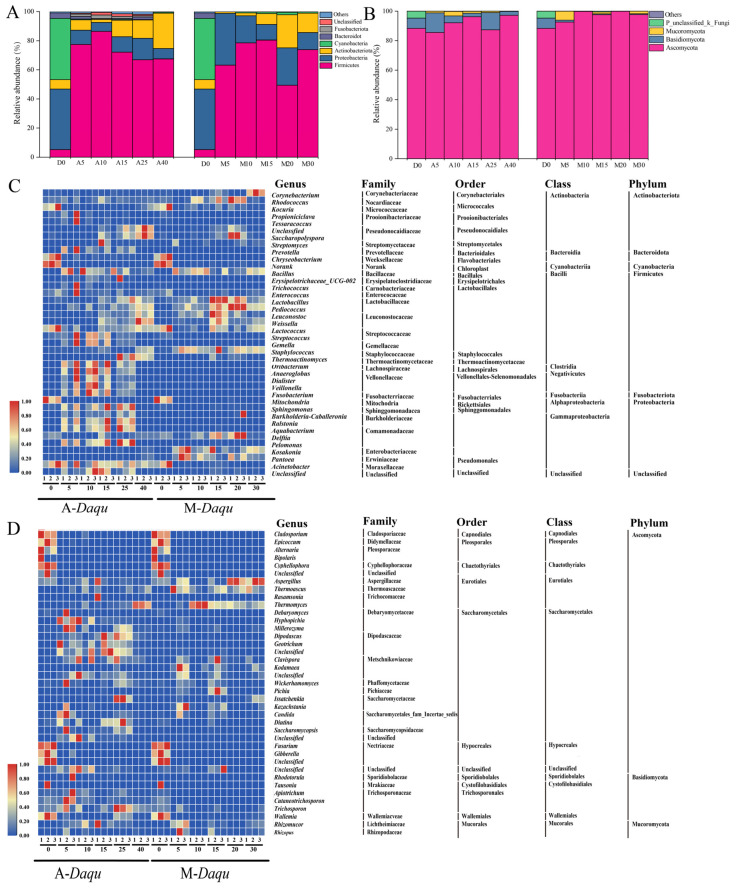
Changes in bacterial (**A**,**C**) and fungal (**B**,**D**) communities in two types of *daqu* during fermentation. (**A**,**B**) Phylum level; (**C**,**D**) genus level. A-*Daqu* and M-*Daqu* represent artificially pressed and mechanically pressed *daqu*, respectively.

**Figure 3 foods-12-02603-f003:**
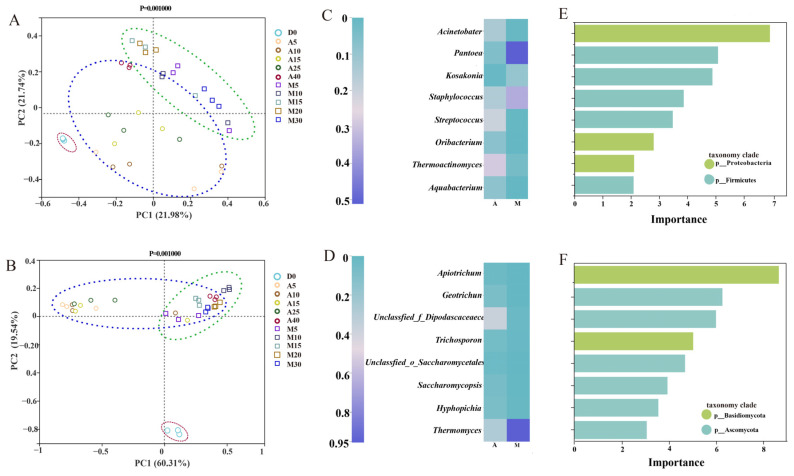
The principal coordinates analysis (PCoA) (**A**,**B**), biomarkers discrimination (**C**,**D**) of the microbial community in two types of *daqu* samples, and the significance values distribution (**E**,**F**) of biomarkers. Dotted circle represent grouped ellipses.PCoA was performed based on bacterial community (**A**) and fungal community (**B**) at genus level. As for biomarker analysis, 8 bacterial genera (**C**) and 8 fungal genera (**D**) that distinguish A-*Daqu* from M-*Daqu* were presented. The bars represent the significance values for bacterial biomarkers (**E**) and fungal biomarkers (**F**) estimated by the random forest learning algorithm, and the colours of the bars indicate the classification of the genus at the phylum level.

**Figure 4 foods-12-02603-f004:**
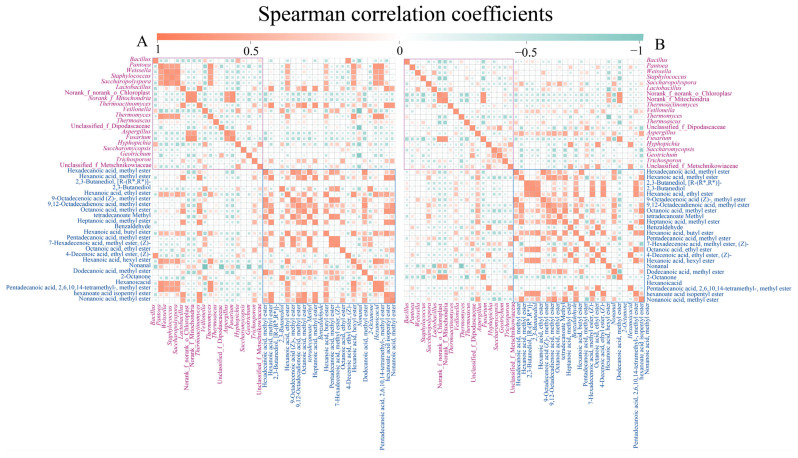
Interaction network of microbial communities and volatile compounds in A-*Daqu* (**A**) and M-*Daqu* (**B**).

**Figure 5 foods-12-02603-f005:**
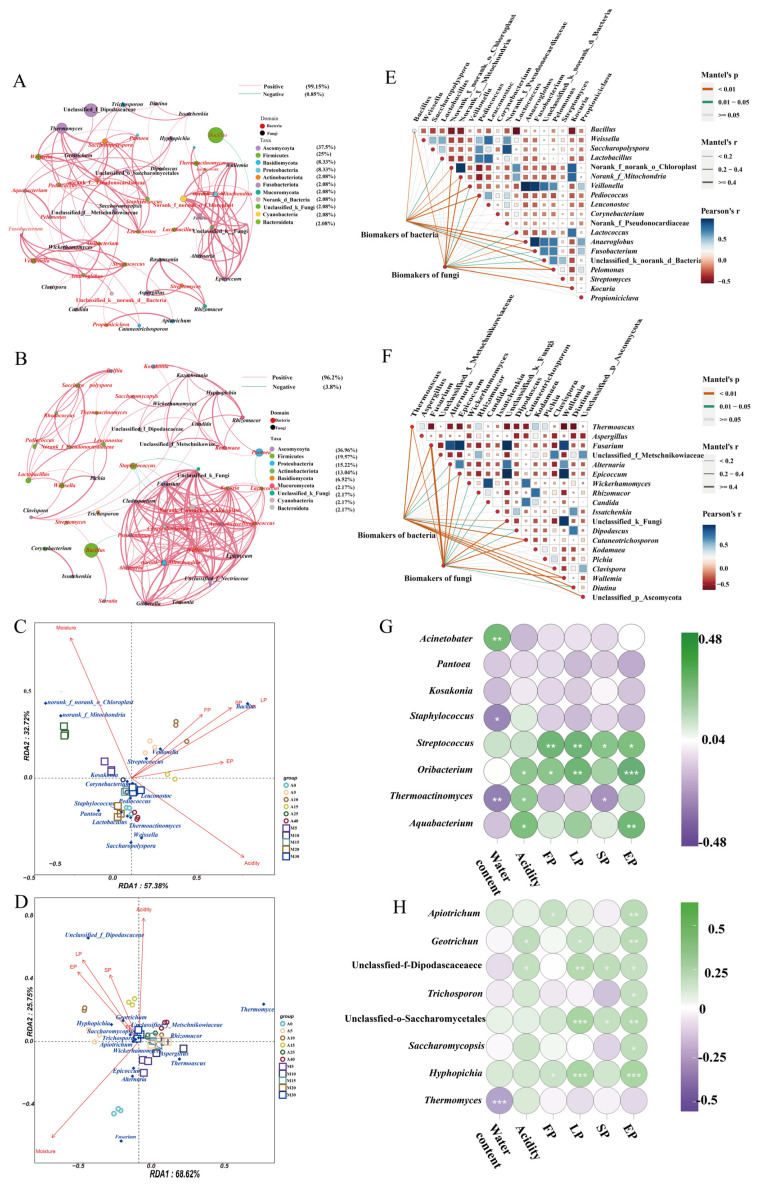
Interaction network of microbial communities and correlation analysis between physicochemical properties and microbial communities. (**A**,**B**) A-*Daqu* and M-*Daqu*; (**C**,**D**) bacteria and fungi. The Mantel test analysis of the biomarkers and each of other microorganisms. (**E**,**F**) bacteria and fungi. Spearman’s correlation between biomarkers and physicochemical properties in two types of *daqu* samples: (**G**,**H**) bacterial and fungal biomarkers. * *p* < 0.05, ** *p* < 0.01, *** *p* < 0.001.

**Figure 6 foods-12-02603-f006:**
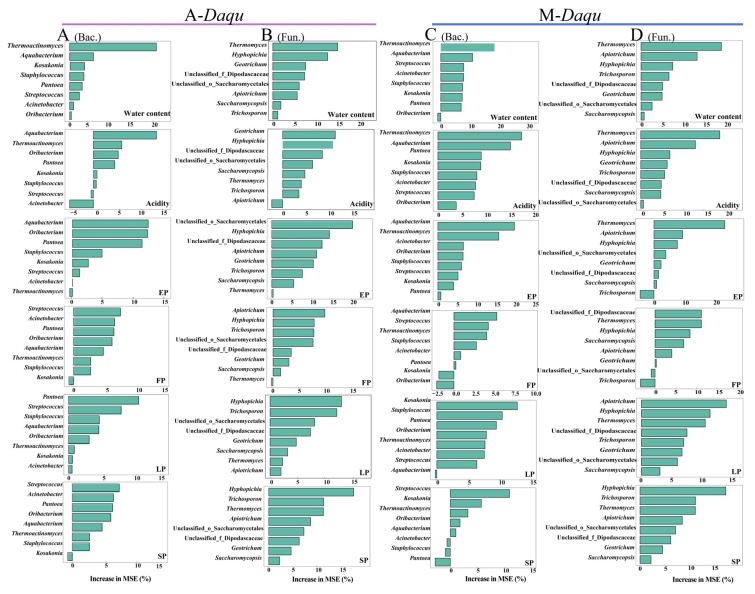
The importance of bacterial (**A**,**C**) and fungal (**B**,**D**) biomarkers as influencing factors for the occurrence of physicochemical property changes in two types of *daqu* during fermentation. (**A**,**B**) A-*Daqu*; (**C**,**D**) M-*Daqu*.

**Figure 7 foods-12-02603-f007:**
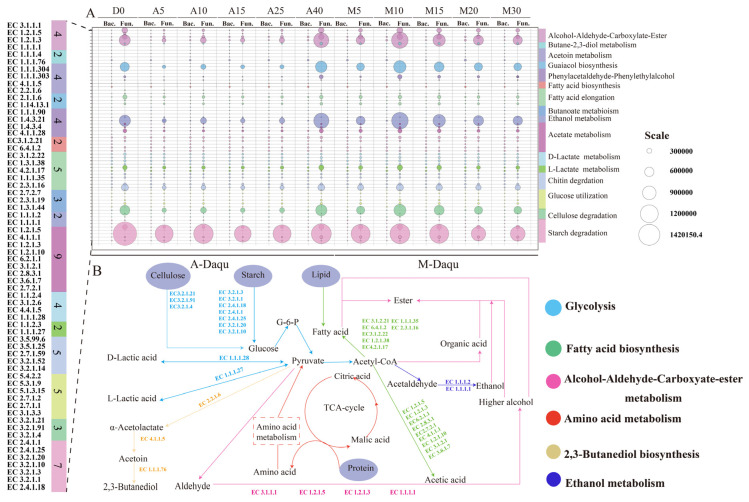
Functional prediction of the microorganisms in two types of *daqu* during fermentation. (**A**) The abundance and functional classification of the main enzymes in the two types of *daqu* during fermentation. (**B**) The primary metabolic pathways involved in the formation of major flavor compounds in the two types of *daqu.*

## Data Availability

Data presented in this study are available upon request from the corresponding author upon reasonable request.

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
