# Peer review of "Exploring the Successions in Microbial Community and Flavor of Daqu during Fermentation Produced by Different Pressing Patterns"

_foods, 2023, doi:10.3390/foods12132603_

Round 1
Reviewer 1 Report
In my opinion, the manuscript entitled Exploring the successions in microbial community and flavor of daqu during fermentation produced by different pressing patterns by Huang et al., is a good one. The introduction provides all the needed information, the materials and methods are enough described and the results are compared with the current state of the art and well discussed.
I just have a small comment:
1. line 199 – before Briefly please put . and not ,
2. lines 125-127 – please use the impersonal pronoun, such as: immediately the fiber was insert.
Thank you!
Reviewer 2 Report
The work under review is scientific and interesting. But I missed the discussion of the results. At the moment, the topic of Daqu starter is actively researched especially from the point of view of the microbial community. I think the authors could have cited their colleagues in the field more in the discussion part, e.g. :
https://doi.org/10.1016/j.ijfoodmicro.2023.110250
https://doi.org/10.1016/j.lwt.2023.114936
https://doi.org/10.1016/j.foodres.2023.113076 and other.
In the summary part, I would like to see conclusions about the technological potential of this study, exactly how it will help improve the quality of daqu and Chinese baijiu. I think this part is not written by the authors.
In addition, the statistical processing of the obtained data is a bit redundant and complicates the interpretation of the results.
Reviewer 3 Report
Line 95: Qu room? If it’s on University campus, please mention it in brackets
Line 97-98: It would be informative if the authors could mention the appropriate temperature conditions even the ambient temperature conditions can be informed
Line 113: Citation should be included for the measurement technique (QB/T 4257-2011).
Line 119: keep period after (Supelco Inc., USA).
Line 125: Change it into passive sentence! Say “Immediately the fiber was inserted into the gas chromatograph……”
Line 113-114: Was the total analysis method for a duration of 15 minutes? Make it clear if there is any shift up or down the temperature after 5 minutes of maintaining the temperature constant at 220 °C
Line 298: Typo, should be corrected as Vulgatibacter , Gluconobacter
(To my knowledge, haven’t come across Vuigatubacter!!) Please explain in case you found this genera from your research studies!
Line 373: Typo, should be corrected as Kosakonia
Line 605: Thermomyces not to be written Thermomyce
Line 628: Any measurements in fermentation time were noticed when observing volatile changes? That would help to keep it as a mark for future experiments when wanted to get a distinctive flavor at a certain point.
Line 659-660: What are the genera which were not involved in enzyme production? Was this noticed in any other literature studies?
Overall comments:
- Methodology could have been explained better, particularly in explaining the analysis of physicochemical and enzymatic parameters.
- If possible, the percentage of acid levels and some other chemical compounds might be well analyzed using chromatographic methods and that would give us an idea of the major dominance in flavor
- References must have been cited more explicitly for statistical analysis methods
Reviewer 4 Report
Dear Editor and Authors,
I send you my review about the article entitled “Exploring the successions in microbial community and flavor of Daqu during fermentation produced by different pressing patterns”.
The aim of the paper, as reported in the scope was to study the succession of microbial communities and volatile compounds of the two types of Daqu during fermentation process.
In my opinion, although the Article is well written, in a good English language and it is well structured, it show, also, some lacks that I reported below.
The introduction is well written and provide to explain the originality of the research, but it should be reported the quantity of Daqu produced and it economic relevance.
Moreover, It should better explained the difference among the M-Daqu and A-Daqu.
In particular should reported what the Authors mean by "mechanical" and "artificial".
The paragraph materials and methods result complete, but it should better described the two different method used to produce M-Daqu and A-Daqu.
Moreover, it should, also, reported the number of trial performed, the number of samples collected in each trial and the number of replicates of analysis performed on each sample.
The results is well presented and they are well discussed, also in comparison to the data reported in the literature.
Finally, the conclusions are present, but should be placed in a special chapter, to this aim I suggest to the authors to use the text from line 661 to line 667 to create a chapter of conclusions.
Best regards
Round 2
Reviewer 2 Report
The authors have taken into account the comments and made changes in the text of the article.